# Efficiency of Managerial Work and Performance of Managers: Time Management Point of View

**DOI:** 10.3390/bs11120166

**Published:** 2021-11-30

**Authors:** Zuzana Lušňáková, Silvia Dicsérová, Mária Šajbidorová

**Affiliations:** 1Faculty of Economics and Management, Slovak University of Agriculture in Nitra, Tr. A. Hlinku 2, 949 76 Nitra, Slovakia; maria.sajbidorova@uniag.sk; 2Faculty of Horticulture and Landscape Engineering, Slovak University of Agriculture in Nitra, Tulipánová 7, 949 76 Nitra, Slovakia; silvia.dicserova@uniag.sk

**Keywords:** efficiency, enterprises, managerial work, managers, time management

## Abstract

Appropriate time management allows individuals to achieve work and personal goals, plan tasks, set priorities, eliminate disruptive effects, and increase work efficiency and productivity. The aim of the paper is to evaluate the effectiveness of managerial work and the performance of managers of food companies in the V4 countries (Slovakia, Czech Republic, Hungary, and Poland) from the perspective of time management principles, point out the shortcomings and reserves that can ensure time management, and propose solutions to improve business practice. We set five research assumptions in order to evaluate the situation comprehensively. A survey carried out from September 2020 to January 2021 involved 1588 managers working at various levels of management. Statistical methods and tests were used for data processing and their subsequent evaluation. The data were processed using Microsoft Excel 2016, the statistical software SAS Enterprise Guide 7.1, and XLSTAT. The analysis showed that three-quarters of managers are aware of the value of their time. More than half of the managers involved try to regularly review their agenda in order to identify gaps in the use of time and to avoid repeating unproductive practices. Only half of the managers make arrangements not to think about work in their free time. The managers spend the most time in their work dealing with administration. Intensifying the implementation of ICT (information and communication technologies) in the work of the manager has the effect of increasing the efficiency of the division and use of working time of managers. Based on our findings, we consider the goal orientation, positive motivation, systematic training, and development of managers as a key prerequisite for efficiency of managerial work and performance of managers and their effective time management.

## 1. Introduction

Tracy lists four “D” of efficiencies. Desire—one must have a strong desire to get one’s time under control and achieve maximum efficiency. Decisiveness—one must make a clear decision to practice good time management techniques until they become his habits. Determination—one must be willing to persevere in the face of all temptations until one becomes an effective time manager. Desire strengthens determination. Discipline—it is the most important key to success in life. One must be disciplined to make time management a lifelong practice. An effective discipline is a willingness to force yourself to pay a price, to do what you know you should do, when you should do it, whether you like it or not. This is critical to success [1].

In the framework of practical coordinating activities for increasing an organization’s overall activity efficacy, the following is taken into account: formal communication channels suitable for relationships in the organizational structure in order to operatively and effectively coordinate all the activities. Implementing coordination activities in good conditions gets result in increasing an activity’s efficacy and efficiency within the company. For increasing managerial activity efficiency, one practices a policy consisting of production or supplying services costs’ continuous discount [2]. 

The efficiency of a manager’s work is influenced by many factors [3]. Skanski has determined that there is a significant interdependence between the degree of management work efficiency (measured using the adjusted Mott’s technique) and leadership styles of managers (measured using Likert scale); the closer the leadership style is to System 4, that is participational, the higher the managerial efficiency is [4]. 

The research of O’Gorman et.al. provides evidence that organizational size is an important determinant of the nature of managerial work. Running a large global company is an exceedingly complex job. The scope of the organization’s managerial work is vast, encompassing functional agendas, business unit agendas, multiple organizational levels, and a myriad of external issues [5]. Unlike any other executive, the CEO has to engage with all stakeholders. However, they, more than anyone else in the organization, confront an acute scarcity of one resource. That resource is time. There is never enough time to do everything that a CEO is responsible for. Despite this, CEOs remain accountable for all the work of their organizations [6]. 

CEOs combine the reality of what they ought to do in the role with who they are as human beings [7]. Excellent CEOs systematically prioritize, proactively schedule, and use interactions with their companies’ important external stakeholders to motivate action. The best CEOs also teach their office staff to help manage the CEO’s energy as thoughtfully as their time, sequencing activities to prevent “energy troughs” and scheduling intervals for recovery practices (for example, time with family and friends, exercise, reading, and spirituality). Doing so ensures that CEOs set a pace they can sustain for a marathon-length effort, rather than burn out by sprinting over and over [8].

In companies, it is important that multiple tasks are managed simultaneously with intelligence and logic, and that tasks are performed sequentially, leading to an efficient division of tasks. In fact, it is not possible for people with the same level of interest to focus on many things at once. Although people focus on only one task, simple ones can follow, which are less important and require little attention. In this context, the consequence of a leadership approach for leaders is that leaders rank their tasks according to importance and carry out the relevant activities gradually [9]. 

The way people perceive time is interesting in itself but also as a predictor of social, cognitive, and affective aspects of behavior. It is also a correlate of important psychological traits [10].

Based on these findings, the research question is formulated as follows: What are the main factors influencing the efficiency of managerial work and the performance of managers from time management point of view which must be improved?

### Stating the Problem

Human capital and its efficient work are an indispensable part of business activities in terms of achieving goals and sustainability. Due to the fact that managerial work is relevant for all strategic systems and processes in companies, it is important to address its effectiveness. By examining this issue among managers in companies in different sectors or countries, our scientific goal is to identify related factors, identify differences and similarities, problems and shortcomings, and make recommendations for improving the current situation in the interests of sustainability.

Effective time management improves productivity, facilitates planning, ensures that tasks are performed at the highest level, helps to set priorities and achieve important tasks, and guides you to achieve set goals [11]. 

Time management skills, technological preparedness, and computer literacy are the basic qualities for the practitioners [12]. Time management and strategic planning strategies are key to succeed [13]. Claessens et al. defined time management as a behavior that aims to achieve the efficient use of time in performing certain goal-oriented activities. Generally speaking, time management refers to activities that involve the efficient use of time, which should lead to increased productivity and reduced stress [14].

Currently, there is great interest in better understanding how time management can help combat time scarcity. People are constantly struggling with the desire to do many things with too limited a resource. This has led to an increased assessment of productivity and busyness [15]. Usually, the ultimate goal in time management is to maximize activity—as many activities, tasks as possible. The second, but perhaps more important, goal is to maximize results. Recent research suggests that strategies that help maximize activity can threaten to maximize results both for work and for leisure [16,17].

Productivity is the time an employee actively spends in the work in which he or she was employed to achieve the desired results expected from an employee’s job description [18]. The findings of Farooq and Sultana provide support for the negative relationship between work from home and employee productivity [19]. The golden rule of productivity is this: the best way to achieve the most is to choose one task to work on. Only focus on this task, without distractions or multitasking. Turn off anything that might disturb you and work on this task until it is done [20]. Increasing the productivity of managers as people who have the knowledge and skills required of each of them can be achieved through proper time management [21]. Keeping the workplace tidy is considered one of the most important tools of time management.

One of the successful entrepreneurs said that the key to his success was “always work from a clean table”. A tidy and organized work environment allows you to work more, in higher quality, in less time, and with better concentration. This means that an organized work environment increases both efficiency and productivity [1]. 

Another tool for managing time management is training. As stated by Van Eerde one month after undergoing time management training, trainees reported a significant decrease in avoidance behavior and worry and an increase in their ability to manage time. The results suggest that time management training is helpful in lessening worry and procrastination at work [22]. The work of Hong et al. dealt with the idea that academic procrastination is negatively related to time management as well as time management being negatively related to learning ineffectiveness [23].

Effectiveness in leading can be identified through efficient time management [24]. Effective time management in the work of a manager is an important prerequisite for achieving partial goals as well as strategic goals of the organization. Within the issue of time management, we can define three categories of people. The view of the first group is that time management is needed to succeed. The second group considers time management unnecessary, and the third group would like to change something in life but has a problem concentrating and then persevering [25].

Work performance is associated with the ability of employees to know set goals, meet expectations, and achieve goals, or perform a standard set of tasks for the organization. Work performance is directly related to the efficiency of employees [26]. Urban describes personal effectiveness as the ratio between the results achieved and the time spent on them. Alternatively, it is the ratio between the time devoted to “real work” and the total time of work or time spent at work. Due to low personal efficiency, there is a significant number of hours we spend on work. If we come to the conclusion that the coefficient of personal effectiveness is lower than 25%, then the person should begin to seriously consider improving his personal effectiveness [27]. The methods and techniques used in planning and organizing one’s own work form an important component of a manager’s professional competence. Effective technology should make it possible to eliminate unnecessary work, and thus shorten the loss of time, reduce stress, reconcile work and private life, increase personal productivity, and work with greater self-realization [28]. 

Skorková states in her publication that up to 66% of managers take their work home (in the evening or on the weekend) and 41% of managers do not feel a balance between private and professional life. The work–life balance theme is implemented in their procedures by many successful companies, which have understood that a satisfied employee is much more efficient and productive. Reconciling the work and private life of a manager is an important measure to ensure the rational setting of goals, the fulfillment of tasks, the fight against disruptive factors, and ensuring work efficiency [29]. Poulose and Sudarsan describe work–life balance as the harmonization of work and non-work life, trying to penetrate into every area of each employee’s life. Work–life balance either improves the conditions for the employees themselves or improves the productivity of the company [30]. According to a 2015 survey of small and medium-sized companies in 16 European countries, in a sample of 5500 respondents, up to a third of employees in Europe admitted that they do not have enough time for their personal lives and almost a quarter say that it is difficult for them to create space for personal life at all. Workplace relationships and management styles have a pervasive impact on the quality work life (QWL), not only mitigating the adverse effects of ICT, but also promoting sustainable productivity and employee performance [31].

Based on the above, these research assumptions were formulated:

P1: We assume that managers who are aware of the value of their time regularly review their agenda to identify gaps in the way they use their time and regularly apply habits that allow them to be more effective in managing their time.

P2: We assume that there are activities in the work of managers that take them more time than others.

Information and communication technologies (ICT) are an essential part of working life for a large number of managers—they significantly reduce the time needed to gather information, communication, administration, or other processes [21]. ICT is being recognized as a major enabler of innovation and new business models, which have the potential to have a major impact on Western economies and jobs. Due to the rapid advances in communication and information technology, ICT usage continues to increase and be-come more pervasive in employees’ daily lives. ICT use can help to improve employees’ productivity [32,33], but it can also lead to increasingly permeable boundaries between work and family domains. 

ICTs may infect the logic of professionalism that takes pride in the quality of the work performance with a dominant managerial logic that places an emphasis on meeting management objectives [34]. Organizations call for employees’ autonomy and self-directedness executed by agile operations and low hierarchies, where learning is also increasingly the responsibility of the individuals and teams themselves and occurs in practice without strong control of the organization [35].

Based on the above, this research assumption was formulated:

P3: We assume that the intensive use of ICT resources has a positive effect on the more efficient use and distribution of a manager’s working time.

The performance of a manager also varies depending on whether he is in a phase when he is full of energy or in a phase when he feels exhausted, tired [36]. Each person has their own performance curve, which is influenced by age, habits, and mental and physical fitness. The elaboration of one’s own performance curve and the use of natural laws in the organization of the working day will clearly guarantee higher performance without extraordinary measures or changes [37].

Each person has an individual performance curve. The performance peak is between 10 a.m. and 11 a.m., and this peak can only be reached once during the day. The first attenuation occurs around 2 p.m. In the evening, the performance curve increases slightly, but after 10 p.m. it begins to decline to the lowest point of performance, which it reaches at about 4 a.m. Based on the description of this curve, the most important and challenging tasks should be performed in the morning. Routine tasks should be performed during the downturn, and more important activities can be revisited after the ascension [38].

There are generally two basic types in relation to human performance: the morning type and the evening type. Neither of these two basic types works better or worse than the other—but they work differently. Knowledge of one’s own personal performance curve helps to better divide work and achieve better efficiency. The early type gets up easily in the morning and is able to get to work fresh and concentrated shortly after waking up. For the morning type, it is advisable to make the most of early morning and afternoon focusing on the most difficult and important tasks that require full concentration and performance [39] (pp. 47–48). 

The evening type has the most energy in the evening, it is difficult for him to get up, shortly after waking up he is not able to start working. The most difficult and important tasks should be performed during the late afternoon until the evening, because the evening type is most productive between 8 p.m. and 11 p.m. [39] (p. 49).

In order to make the manager’s work more efficient, it is also necessary to take into account the disturbance curve. This curve is typical of a normal weekday in the office. Various types of disturbances, such as phone calls, unannounced visits, surprising meetings, colleagues who want to take over something, etc. are kicked out at certain times of the day. The disturbance curve should be taken into account by each manager when planning his working time on a daily basis [39] (p. 49).

Based on the above, these research assumptions were formulated:

P4: We assume that managers are most productive between 9 a.m. and 12 p.m.

P5: We assume that managers are most disturbed at the same time, between 9 a.m. and 12 p.m.

## 2. Materials and Methods

The aim of this paper is to evaluate the effectiveness of managerial work and the performance of managers of food companies in the V4 countries (Slovakia, Czech Republic, Hungary, and Poland) from the perspective of time management principles.

The research was carried out among managers of food companies operating in all V4 countries. Food businesses are an important part of the national economy not only of the V4 countries and they make a significant contribution to the production of food and beverages and also provide jobs for the population.

In order to obtain primary qualitative and quantitative data on the productivity and work efficiency of business managers, a questionnaire was compiled and a survey was conducted. The survey involved 1588 valid respondents—managers working at various levels of management in food businesses in the V4 countries. The questionnaire survey was conducted online from September 2020 to January 2021. The return rate of the questionnaire was 97%, as its implementation was communicated in advance and the addressed managers agreed to participate in the survey.

The questionnaire was based on the test “How do you manage your time” by Godefroy and Clark [40]. In the questionnaire, we applied the Likert scale in selected questions, which is considered to be one of the most used and most reliable techniques for measuring attitudes in questionnaires. The respondent can answer individual statements using a five-point scale, where 1 represents “strongly disagree” and 5 represents “strongly agree”. Thanks to the application of the Likert scale also used by Sellitto, Camfield, and Buzuku [41], we obtain the degree of the respondent’s agreement with the given statement. However, this scale allows us to determine not only the content of the attitude, but also its approximate strength [42]. 

The structure of the questionnaire with the selected types of questions in their full text is shown in the Table 1.

Statistical methods and tests were used for data processing and their subsequent evaluation. The data were processed using the Microsoft Excel 2016 spreadsheet into clear tables and graphs, as well as through the statistical software SAS Enterprise Guide 7.1 and XLSTAT. 

According to Prokeinová, from a methodological point of view, it is very difficult to apply any statistical method to this type of ordinal data. Through the application of basic descriptive characteristics, we can obtain values of mode, median, and averages of individual preferences of respondents [43]. The Cronbach’s alpha coefficient, which was applied, is one of the methods for evaluating the reliability of scales and is one of the most widely used [44] (p. 189). Correlation analysis is an applied statistical procedure that describes the relationship between numerical variables. The correlation coefficient is used to measure the two-sided linear dependence of two variables [44] (pp. 175–176).

The most frequently used rank correlation coefficient—Spearman’s coefficient—was also used [45] (pp. 69–70). The values of the Spearman correlation coefficient were based on the interpretation according to De Vaus [46]. Interpretations of the individual values are shown in Table 2 below: 

The Mann–Whitney test is used to verify the difference in order averages [45]. Lyócsa, Baumöhl, and Výrost state that by extending the Mann–Whitney test for two independent samples to k samples, we get to the Kruskal–Wallis test [47]. The established null hypothesis H0 states that there is no difference in the mean values of the examined groups (e.g., in the opinions of respondents within one question). Subsequently, the alternative hypothesis H1 will assume that there is no difference in the mean values between the examined files [48] (pp. 321–323). The Friedman test is used to verify whether the level of the monitored trait depends or does not depend on changing conditions. It can be applied for nominal or ordinal measurement scale [49] (p. 77). The McNemar test is used to analyze the agreement of two evaluators. A statistically significant test result means uneven distributions of discrepancies between categories. Thus, the evaluators do not agree more in some categories than in others [45] (pp. 174–175).

## 3. Results

A questionnaire survey was prepared and carried out for the purpose of obtaining primary qualitative data. The target group of respondents were managers of food companies in the V4 countries (Slovakia, Czech Republic, Poland, and Hungary), operating at various levels of management. A total of 1588 respondents took part in the questionnaire survey. Of them, 50.74% were men and 49.26% were women. The largest group of managers was 36.67%, aged 26–35, and the second largest group of 26.67% was aged 36–45. This was followed by a group of managers aged 46–55 (17.40%), under 25 (10.74%), 56–65 (7.78%), and the smallest representation in the research sample was managers over 65.

In terms of the level of management, the group of managers working at the lowest level of management had the largest representation (44.82%). Followed by middle management (33.33%) and top management (21.85%). Most respondents (27.78%) have been working in the company for 11 or more years. In the sample, 26.67% of managers have been working in the company for 2 to 3 years, 22.22% of respondents for 6 to 10 years, and at least (7.03%) of respondents have been working in the company for less than 1 year.

Given that a substantial part of our questionnaire consisted of statements in which respondents had to express the degree of their agreement or disagreement on a 5-point Likert scale, it was necessary to test the internal consistency of the scale used in the questionnaire. We tested the entire questionnaire in a single, unique check. The value of Cronbach’s Alpha is found in Table 3.

The Cronbach’s Alpha coefficient reached 0.894325, which represents a sufficient consistency of the scale used in the questionnaire. Based on the above, a further analysis was performed. 

Effective time management improves productivity, facilitates planning, ensures the fulfillment of tasks at the highest level, and helps with setting priorities, fulfilling important tasks, and achieving set goals [11]. We identified five statements whose average and most common values expressed by mode are shown in Figure 1.

As we described in the theoretical part, personal effectiveness can be understood as the ratio between the results achieved and the time we spent on them. That is why the statement E1 was formulated “I regularly review my agenda to identify gaps in the use of my time and to avoid repetition unproductive practices”. Respondents’ answers reached an average value of 3.57 and the most frequent value was 4 (I partially agree). More than half (56.66%) of managers try to increase their personal efficiency. Of the managers, show significant reserves in personal efficiency.

If the manager is aware of the value of his time, he can better manage its use. The statement E2 “I am aware of the value of my time” achieved an average response of 4.11 and the most frequent response was 5 (I certainly agree). Three quarters (75.55%) of managers are aware of the value of their time.

The E3 statement “I regularly apply habits that allow me to be more effective in managing my time” reached an average of 3.89 and the most common response at level 4 (I partially agree). More than two-thirds (69.63%) of managers regularly apply habits that allow them to be more efficient in managing their time. The results showed 7.03% of managers do not apply such habits.

It is very important to maintain a certain balance between work and leisure. It usually happens that managers are unable to relax even in their free time. The statement E4 “I make arrangements so that I do not think about work in my free time” reached an average level of 3.68 and the most common answer was 5 (I definitely agree). At present, with an increasing emphasis on work–life balance, up to a fifth (20.73%) of managers do not make arrangements not to think about work in their free time.

The E5 statement “I am the owner of my time” had an average value of 3.61 and the most frequent answer was answer 4 (I partially agree). Almost two thirds (60.74%) of managers consider themselves owners of their time.

In statements dealing with managerial efficiency, we found the existence of statistically significant differences between managers’ responses due to their gender, age, position in the firm, and length of time in the company using the Mann–Whitney and Kruskal–Wallis test. The results are shown in Table 4. Values marked “*” represent statistically significant differences at the 0.05 significance level and values marked “**” represent statistically significant differences at the 0.01 significance level.

In the case of statement E1 “I regularly review my agenda to identify gaps in the use of my time and to avoid repetition of unproductive practices”, statistically significant differences were found in the responses of managers only with regard to the length of time in the company, at the level significance 0.05. No statistically significant differences were found to be due to gender, age, and position in the firm in the responses of managers participating in our research. Managers in the company for less than one year do not regularly review their agenda to identify gaps in the use of their time and to avoid repeating unproductive practices.

In the E2 statement “I am aware of the value of my time”, there are highly statistically significant differences in the answers of managers with regard to their gender and position in the firm. More men than women agreed with this statement. Most managers at the lower management level stated that they are not aware of the monetary value of their time.

In the E3 statement, “I regularly apply habits that allow me to be more effective in managing my time,” there were no statistically significant differences in managers’ responses with respect to any identifying feature. This means that it does not matter what gender or age the manager is, it also does not matter what position in the firm he/she has and how long he has been working in a particular company. None of these factors affect the application of habits that allow managers to be more efficient in managing their time.

In the case of the statement E4 “I make arrangements not to think about work in my free time”, highly statistically significant differences were found in the answers of managers with respect to the position in the firm at the significance level 0.01 and statistically significant differences between managers with regard to their length of time in the company at the level of significance 0.05. The largest share of managers, for whom it is a challenge not to think about work in their free time, are managers at the highest level of management and managers who have been working in the company for more than 6 years.

In the E5 statement “I am the owner of my time”, highly statistically significant differences were found only with respect to the position in the firm. The largest share of managers who do not consider themselves the owners of their time are managers in top management.

Subsequently, we examined the existence of dependencies between individual statements regarding the effectiveness of managers’ work with each other. The values of the Spearman correlation coefficient can be found in Table 5. Values marked “**” represent highly statistically significant correlations at the significance level of 0.01, and those marked “*” represent statistically significant correlations at the significance level of 0.05, which belong to the category of medium to significantly strong correlations.

The strongest dependence in Table 5 is between statements E1 and E3, it has a value of 0.38224. It represents a medium to significantly strong correlation between regular review of the agenda and regular application of habits that allow managers to be more efficient in managing their time. Managers who regularly review their agendas to identify gaps in the way they use their time and to avoid repeating unproductive practices regularly apply habits that allow them to be more efficient in managing their time.

The second highest correlation is between the statements E2 and E3 and belongs to the medium to significant dependencies. Managers who are aware of the monetary value of their time regularly apply habits that allow them to be more efficient in managing their time.

Research premise P1 “We assume that managers who are aware of the value of their time regularly review their agenda to identify gaps in the use of their time and regularly apply habits that allow them to be more efficient in managing their time” based on Spearman’s correlation findings we confirmed.

A medium to substantial dependence was found between the statements E1 and E2. Managers who are aware of the value of their time regularly review their agenda to identify gaps in the way they use their time and to avoid repeating unproductive practices.

To determine the existence of statistically significant differences between the activities, the implementation of which takes managers the most time, we applied the Friedman test, the result of which can be found in Table 6.

Based on the results of the application of the Friedman test, we can state that there are statistically significant differences between the activities that managers perform during the day. We state that there are activities that managers deal with during the day longer than others. The applied Nemenyi method offers us a more comprehensive view of the evaluation of which activities take the most time for managers. The result of the multipair comparison can be found in Table 7.

Based on Nemenyi’s method, we found that the administration (included in group A) takes the managers the most time in their work. They take the least time for managers to act as delegating tasks to subordinates and catching up on what they missed (included in group D). The result of Nemenyi’s method is documented by the Demshar plot in Figure 2.

We confirmed research premise P2 “We assume that there are activities in the work of managers that take them more time than others.” based on the Friedman test.

The increasing use of ICT resources in the work of the manager also affects the efficiency of his work. One of the factors positively influencing the efficiency of time management is the intensive use of ICT resources. To determine the impact of more intensive implementation of ICT in the work of the manager, we applied the McNemar test, the results of which can be found in Table 8.

Based on the McNemar test, we can say that intensifying the implementation of ICT in the work of the manager has had the effect of increasing the efficiency of the distribution and use of working time. The research premise P3 “We assume that the intensive use of ICT resources has a positive effect on the more efficient use and distribution of manager’s working time” was therefore confirmed.

Next, we were interested in the performance curve and the disturbance curve and part of the issue of efficiency in the work of the manager. Figure 3 shows the summary results of the responses of the managers participating in our research (in percentage terms). Shown are parts of the day that the managers identified as the time when they are most productive.

Most (59.63%) respondents consider that they are the most productive with respect to achieving the highest performance in the time from 9 a.m. to 12 p.m. Of course, each manager is an individual personality and everyone can achieve the highest performance in different parts of the day. According to several authors (Burger, 2012; Knoblauch-Wöltje, 2006) and also according to our research, this part of the day (9 a.m. to 12 p.m.) is the most productive part of the day. Equally, 20% of managers participating in the research achieve the highest performance from 6 a.m. to 9 a.m.

Productivity at work and, of course, the efficiency of performing individual tasks and achieving set goals are greatly influenced by various forms of disruption at work. Managers can be interrupted by various phone calls, e-mails, their colleagues, unannounced visits, unexpected meetings, or otherwise. The respondents answered the question “At what part of the day are you most disturbed from your point of view?” and the result in percentage terms is shown in Figure 4.

The majority of respondents (43.70%) are most disturbed in the time from 9 a.m. to 12 p.m., which again confirmed the views of individual authors [36,37,38,39] regarding disturbance curves. Between 12 p.m. and 3 p.m., 22.96% of respondents are the most disturbed. However, it is also interesting to find that 16.67% of managers surveyed indicated that it is not possible to determine in which part of the day they are most disturbed.

Through the analysis, we found that more than half of the managers involved in the survey try to regularly review their agenda to identify gaps in the use of time and to avoid repetition of unproductive practices. Three quarters of managers are aware of the monetary value of their time and regularly apply habits that allow them to be more efficient in managing their time. Less than two thirds of managers take steps not to think about work in their free time. Almost two thirds of managers consider themselves masters of their time. The administrators spend the most time in their work in the administration. They take the least time for managers to act as delegating tasks to subordinates and catching up on what they missed. Intensifying the implementation of ICT in the work of the manager has an impact on increasing the efficiency of the distribution and use of working time of managers. Most managers achieve the highest performance from 9 a.m. to 12 p.m. and at the same time, most managers feel disturbed.

## 4. Discussion

Through an analysis of the issues of efficiency of managerial work and performance of managers of food companies in the V4 countries, we found that more than half of managers try to increase their personal efficiency, three quarters of managers are aware of the value of their time, more than two thirds of managers regularly have habits that allow them to be more efficient in managing their time, one fifth of managers do not take measures, so they think about work in their free time, two fifths of managers do not consider themselves like owners of their time, the length of time in the company has an impact on the regular reassessment of the manager’s agenda (managers working in the company for less than one year do not regularly review their agenda to identify gaps in the use of their time and to avoid repetition of unproductive practices), gender and position in firm have a proven impact on awareness of the value of your time (more women than men and lower-level managers said they were unaware of the monetary value of their time), none of the identifying factors of managers has an impact on the regular application of habits that allow managers to be more effective in managing their time, position in the firm and the length of time in the company have an impact on the implementation of arrangement so that managers do not think about work in their free time (managers at the highest level of management and managers working in the company for more than six years do not take arrangements so that they do not think about work in their free time), position in the firm has an impact on considering oneself the masters of one’s time (top managers do not consider themselves like owners of their time), administration takes the most time from managers at work, delegating tasks to subordinates and catching up on what they did not manage, in contrast, takes the least amount of their time, the introduction of an electronic diary (whether on a computer, tablet or mobile phone) has had the effect of increasing the efficiency of the distribution and use of working time.

Time should be understood in relation to others, not as a private matter. Time, including its use and experience, is subject to negotiations, power relations, and inequality [50]. Intellectually productive people usually have more things that they would like to do, or need to do, than they have time [51]. Time estimation increases the perceived control of time, which in turn decreases stress response. However, taking each moment as it comes reduces perceived control of time, which in turn increases stress response [52]. Self-management, especially regarding the temporal dimension of work, has become a critical issue. This is exemplified by the increased demand for time management training in managerial practice [53,54]. The research done by Claessens et al. showedn that perceived control of time is an important variable when studying the effects of planning behavior on the one hand, and the effects of job characteristics on the other [55]. 

Therefore, what are the main factors influencing the efficiency of managerial work and performance of managers from time management point of view which must be improved? Based on our research, we consider it important to state that goal orientation, positive motivation, systematic training, and development of managers are key prerequisites for effective time management. Nevertheless, Macan found a weak relationship between participation in time management training and one facet of time management behavior, i.e., ‘setting goals and priorities,’ also referred to as planning behavior [53].

Every manager should follow the rules of Šuleř in order to get his time under control. Always make sure you know the priority of the tasks—you only need to devote as much time to each task as is appropriate to its importance. Identify and eliminate “self-disturbance”—notice when and on what occasions you are unable to concentrate, when thoughts run away from work, or you tend to deal with issues unrelated to your work. Reduce time spent on unnecessary discussions—regular communication with your surroundings is important, but it should be managed and should not disrupt the manager’s work plan. Keep a record of disruptions—keep a record of all disruptions in one day, indicating the duration, type, reason, and usefulness. Regularly analyze the causes of disturbance—analyze the record and focus on eliminating the largest consumers. You should only allow 5 to 10% of interruptions, which are of course necessary and to some extent useful for you. Avoid unexpected visits—you should not accept anyone who is not announced in advance. Exceptions should be isolated and under prespecified conditions. Compromise with your supervisor—it may be that your supervisor is the biggest consumer of your time. In this case, it is necessary to discuss the situation with him or involve him in planning your time, which will reduce the amount of wasted time. Do not be a slave to your subordinates—by delegation you can fall into the trap of having your subordinates turn to you again with delegated tasks. Learn to say no—we should be as positive as possible, but that does not mean you can’t refuse some requests (especially those that are very time consuming) [56].

Time management skills are something that is an important part of all professions and managers [57]. Despite recommending excellent time management skills in organizations and workspaces, it appears that still a small number of studies have truly documented the empirical soundness of essential time management practices [58]. 

## 5. Conclusions

The research assumption “that managers who are aware of the value of their time regularly review their agenda to identify gaps in the use of their time and regularly apply habits that allow them to be more effective in managing their time” was confirmed from Spearman’s correlation coefficient values.

We found that there is a medium to significant positive dependence between realizing the value of their time and the regularity of reviewing the agenda, in order to uncover gaps in the use of their time and regularly apply habits that allow them to be more effective in managing their time.

The research assumption “that there are activities in the work of managers that take them more time than others” was confirmed on the basis of Friedman’s test. We found that there are activities that managers take during the day longer than others. Based on Nemenyi’s method, we found that the administration takes the most time for managers at work. Activities such as delegating tasks to subordinates and catching up on what they missed take the least time for managers.

The research assumption “that the intensive use of ICT resources has a positive effect on the more efficient use and distribution of manager’s working time” was confirmed on the basis of the McNemar test.

The results of the survey point to the strengths of managing the work of managers, but also to the reserves and shortcomings. Theoretical background to the chosen issue offers procedures for improvement. Based on the above, we recommend time planning and strict adherence to the plan in order to formalize these activities as well as eliminate time thieves. Success in the era of the development of information and communication technologies and the necessary digitization, which is significantly influenced by the current situation of COVID-19, requires highly qualified and competent employees. We recommend the systematic creation of educational programs that will be highly motivating for employees in order to acquire new knowledge and skills. As part of the knowledge management system, we recommend that companies focus support on the process of acquiring, distributing, communicating, and storing knowledge. Continuous systematic and targeted development, training, and education of employees are the key to corporate success. 

In the research, we used several methods of obtaining and evaluating, especially quantitative data. In the interest of a comprehensive study of the issue of the effectiveness of managerial work, it is important to focus attention and apply the methods of qualitative research. Although these are time-consuming and evaluation-intensive, they provide opportunities to obtain the necessary additional information. As part of further research, we see space in the application of several diagnostic methods applied in working with human resources.

Our conclusions are characteristic not only for the food industry. We are currently conducting research in companies from other sectors of the national economy. The findings processed so far have confirmed similar results.

## Figures and Tables

**Figure 1 behavsci-11-00166-f001:**
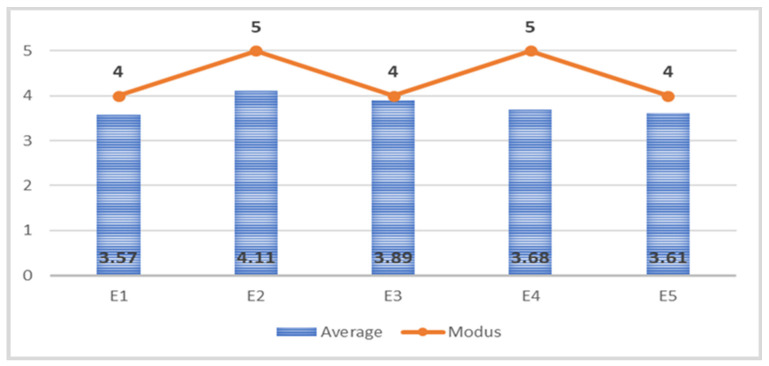
Position characteristics evaluating efficiency. Source: own research, own processing.

**Figure 2 behavsci-11-00166-f002:**
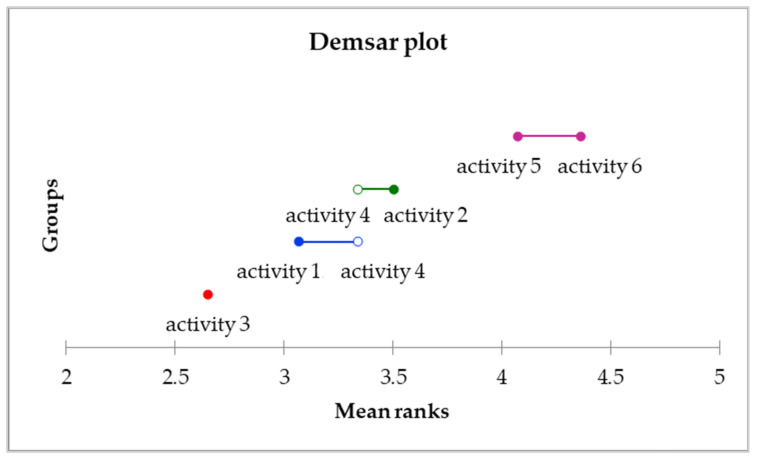
Demshar plot for the most performed activities. Source: own research, own processing.

**Figure 3 behavsci-11-00166-f003:**
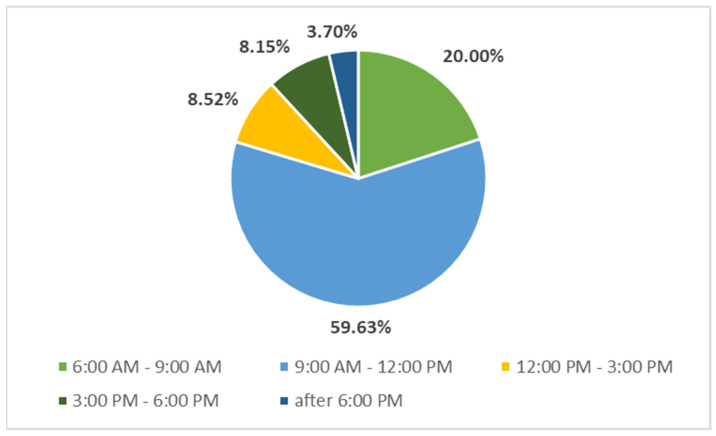
The part of the day with the highest performance. Source: own research, own processing.

**Figure 4 behavsci-11-00166-f004:**
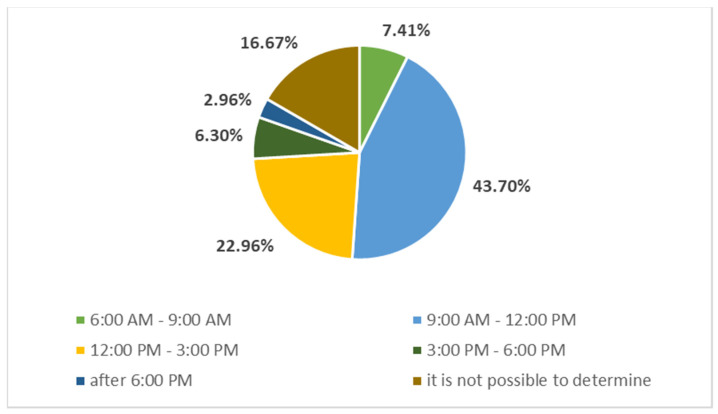
The part of the day with the highest disturbance. Source: own research, own processing.

**Table 1 behavsci-11-00166-t001:** Questionnaire.

Identification Questions
Gender
Your age
What level of management are you currently at?
How many years have you been working for the company?
**Statements with the expression of the agreement/disagreement**
I regularly review my agenda to identify gaps in the use of my time and to avoid repetition unproductive practices. (E1)
I am aware of the value of my time. (E2)
I regularly apply habits that allow me to be more effective in managing my time. (E3)
I make arrangements not to think about work in my free time. (E4)
I am the owner of my time. (E5)
I regularly review my agenda to identify gaps in the use of my time and to avoid repetition unproductive practices. (E6)
**Questions with a choice of options**
Which activity at work takes you the most time?
Which part of the day do you achieve the highest performance, are you the most productive?
At what time of the day are you most disturbed in your view?
**Statements with the possibility of agreement / disagreement (Yes/No)**
In the past, before the introduction of the electronic diary (computer, tablet, mobile phone, …), I often experienced problems with the efficient use of my working time.
My working time is divided and used effectively after the introduction of the electronic diary (computer, tablet, mobile phone, …)

Source: own processing.

**Table 2 behavsci-11-00166-t002:** Interpretation of Spearman correlation coefficient values.

Correlation Value	Interpretation of Dependence
0.01–0.09	Trivial or none
0.10–0.29	Low to medium
0.30–0.49	Medium to substantial
0.50–0.69	Substantial to very strong
0.70–0.89	Very strong
0.90–0.99	Almost perfect

Source: De Vaus (2002), own elaboration.

**Table 3 behavsci-11-00166-t003:** Cronbach’s Alpha coefficient.

Cronbach Coefficient Alpha	
Variables	Alpha
Raw	0.868965
Standardized	0.894325

Source: own research, own processing.

**Table 4 behavsci-11-00166-t004:** Results of the Mann–Whitney test and the Kruskal–Wallis test—efficiency.

	Values of the Mann–Whitney Test and Kruskal–Wallis Test			
	Gender	Age	Position in the Firm	Length of Time in the Company
**E1**	0.4275	0.1178	0.1538	0.0338 *
**E2**	0.0005 **	0.0292	<0.0001 **	0.2010
**E3**	0.0562	0.0520	0.5092	0.5538
**E4**	0.3984	0.1379	0.0018 **	0.0265 *
**E5**	0.4849	0.8244	0.0006 **	0.0577

Source: own research, own processing. Values marked “*” represent statistically significant differences at the 0.05 significance level and values marked “**” represent statistically significant differences at the 0.01 significance level.

**Table 5 behavsci-11-00166-t005:** Spearman correlation coefficient—efficiency.

	E1	E2	E3	E4	E5
**E1**	1.00000	0.35284 **	**0.38224** **	−0.02594	0.17385 **
**E2**	0.35284 **	1.00000	0.34297 **	0.00347	0.28544 **
**E3**	**0.38224** **	0.34297 **	1.00000	0.08366	0.19204 **
**E4**	−0.02594	0.00267	0.08366	1.00000	0.23156 **
**E5**	0.17385 **	0.28544 **	0.19204 **	0.23156 **	1.00000

Source: own research, own processing. Values marked “**” represent highly statistically significant correlations at the significance level of 0.01, and those marked “*” represent statistically significant correlations at the significance level of 0.05, which belong to the category of medium to significantly strong correlations.

**Table 6 behavsci-11-00166-t006:** Friedman test for the most performed activities.

Q (Observed Value)	308.905
Q (Critical value)	12.160
DF	5
*p*-value (one-tailed)	<0.0001
alpha	0.05

Source: own research, own processing.

**Table 7 behavsci-11-00166-t007:** Nemenyi’s method for the most performed activities.

Sample	Average Order	Groups			
administration (activity 3)	2.713	A			
scheduled meetings and conferences (activity 1)	3.148		B		
fulfillment of requests from colleagues (activity 4)	3.381		B	C	
unscheduled meetings and visits (activity 2)	3.521			C	
delegation of tasks to subordinates (activity 5)	4.173				D
catching up on what I missed (action 6)	4.459				D

Source: own research, own processing.

**Table 8 behavsci-11-00166-t008:** McNemar test for the impact of intensifying the implementation of ICT in the work of the manager.

Q	92.542
z (Observed value)	9.247
|z| (Critical value)	1.970
*p*-value (Two-tailed)	<0.0001
Alpha	0.05

Source: own research, own processing.

## Data Availability

Data supporting the reported results can be accessed on request to the main author.

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
