# Peer review of "Efficiency of Managerial Work and Performance of Managers: Time Management Point of View"

_behavsci, 2021, doi:10.3390/bs11120166_

Round 1

Reviewer 1 Report

Abstract: Please explain to the audience of the journal what V4 countries are. Not all readers come from East Europe. The abstract is too large (~400 words). Please try to reduce it without reducing information. You provide sufficient information on objectives, methods, and results, which is excellent, but the text is wordy. Try to tell the same content with at most 300 words. Introduction: It is not a good style to employ lists and figures in the introduction. Please, avoid them replacing by cursive, descriptive text and reallocating the figure to the next section. I believe that – Likert scale – is better than - Likert’s method -. Please replace. Please avoid short paragraphs (two lines) as in lines 105 – 108. Assumptions are well-derived, and you should do the same regarding the research gap and the research question. It would benefit the audience if you formally formulate the research question that your study aimed to answer. In a general manner, your Introduction is too large and clearly confounds two elements of a scientific paper, the presentation, and the problem formulation. I suggest to you split the current Introduction in two sections. The first, named section 1 - Introduction, should finish with the research gap and the research question. The second, named section 2  - Stating the problem (or something similar), in which you derive the assumptions and position your study in front of others (in short, regarding the research gap, what others have already done, what you will do, and what will be left to be done in further research). In the end, a synoptic table showing a snapshot of the papers you reviewed and would be useful to readers interested in using your research in their own research. Materials and Methods: A demographic analysis is due (please see 10.1016/j.spc.2020.04.007, Figure 1, for an example) regarding the companies, not only the respondents. In the same piece (page 96), you will find some data on the return rate of questionnaires (you may wish to cite the article if you find it useful for the audience). The amount of valid respondents is relevant information. Another missing piece of information on the survey is how many waves you perform and what the percentage rate of return of each one is. It is usual to send more than once the questionnaires for those that did not answer the first requirement. From line 224 on, you describe stats procedures. The text is wordy and tedious for an audience of a top journal. I don't believe the audience needs to read about what Cronbach's alpha or correlation is. A list of the steps and tests performed without too many obvious details (perhaps even a table) should satisfy.  You should show the questions in the text. It is very difficult to follow your reasoning without knowing the questions and how they were structured. About Table 2, I believe you have a problem here. It is not clear if you have constructs of questions (a structured questionnaire) or if you tested the entire questionnaire in a single, unique check (please verify the already-mentioned paper to see the so-called constructs and the reliability tests). I believe that 0.9 is a too high alpha. It probably informs that you have multicollinearity in your answers and that you should employ factor analysis to separate the questions in intermediate, correlated blocks, the constructs. It also claims that you possibly have too many questions and some of them are inquiring the same thing under different names.  Please consider processing a little more the questions before proceeding to the results. If you decide to separate the questions in constructs, a discriminant validity test is due. These concerns will not modify your results, which is good, but may reinforce the validity of the survey (reinforce that you found exactly what you were searching for) and the reliability of the findings (another similar application in a different context or timeline is likely to produce similar results). In a further study, I suggest inserting an ultimate question, such as: My company is timely-efficient, and this efficiency allows making more money (or something similar). The ultimate question must embrace some type of consequence or implication of a work-efficiency policy, that is, an indicator that reflects the effects of the strategic actions. Perhaps this question already exists in your questionnaire, but as you did not present it, I cannot propose anything. This ultimate question operates as an endogenous, dependent variable and forms a regression model (y= f(x1, x2,… xn)) with the current exogenous, independent ones. It would greatly increase the impact of your study if such a question exists because it could give a clue on what practice, belief, or behavior is more effective in increasing time productivity. The analysis and discussion is fine. In the last chapter, please avoid lists and mainly lists within lists. This is a closing chapter in which you finish the article and provide a clue on further research. A list of results and a discussion on their implications should be located in the previous chapter. Congrats on your study. Looking forward to receiving your next version.

Author Response

Comments and Suggestions for Authors

Abstract:

Please explain to the audience of the journal what V4 countries are. Not all readers come from East Europe. DONE in Abstract

The abstract is too large (~400 words). Please try to reduce it without reducing information. You provide sufficient information on objectives, methods, and results, which is excellent, but the text is wordy. Try to tell the same content with at most 300 words. DONE

Introduction: It is not a good style to employ lists and figures in the introduction. Please, avoid them replacing by cursive, descriptive text and reallocating the figure to the next section. I believe that – Likert scale – is better than - Likert’s method -. Please replace. DONE

Please avoid short paragraphs (two lines) as in lines 105 – 108. DONE

Assumptions are well-derived, and you should do the same regarding the research gap and the research question. It would benefit the audience if you formally formulate the research question that your study aimed to answer. DONE (lines 84-86)

In a general manner, your Introduction is too large and clearly confounds two elements of a scientific paper, the presentation, and the problem formulation. I suggest to you split the current Introduction in two sections. The first, named section 1 - Introduction, should finish with the research gap and the research question. The second, named section 2  - Stating the problem (or something similar), in which you derive the assumptions and position your study in front of others (in short, regarding the research gap, what others have already done, what you will do, and what will be left to be done in further research). DONE

In the end, a synoptic table showing a snapshot of the papers you reviewed and would be useful to readers interested in using your research in their own research. Materials and Methods: A demographic analysis is due (please see 10.1016/j.spc.2020.04.007, Figure 1, for an example) regarding the companies, not only the respondents. In the same piece (page 96), you will find some data on the return rate of questionnaires (you may wish to cite the article if you find it useful for the audience). As our intention was to examine the effectiveness of the manager's work regardless of belonging to the institution (apart from the definition that we conduct research within food companies) none of the questions examined the identification factors or characteristics of the company in which the manager works.

The amount of valid respondents is relevant information. Another missing piece of information on the survey is how many waves you perform and what the percentage rate of return of each one is. It is usual to send more than once the questionnaires for those that did not answer the first requirement. DONE lines 242-244

From line 224 on, you describe stats procedures. The text is wordy and tedious for an audience of a top journal. I don't believe the audience needs to read about what Cronbach's alpha or correlation is. A list of the steps and tests performed without too many obvious details (perhaps even a table) should satisfy.  You should show the questions in the text. It is very difficult to follow your reasoning without knowing the questions and how they were structured. DONE line 259

About Table 2, I believe you have a problem here. It is not clear if you have constructs of questions (a structured questionnaire) or if you tested the entire questionnaire in a single, unique check (please verify the already-mentioned paper to see the so-called constructs and the reliability tests). Answered in line 320

I believe that 0.9 is a too high alpha. It probably informs that you have multicollinearity in your answers and that you should employ factor analysis to separate the questions in intermediate, correlated blocks, the constructs. It also claims that you possibly have too many questions and some of them are inquiring the same thing under different names.  Please consider processing a little more the questions before proceeding to the results. If you decide to separate the questions in constructs, a discriminant validity test is due. These concerns will not modify your results, which is good, but may reinforce the validity of the survey (reinforce that you found exactly what you were searching for) and the reliability of the findings (another similar application in a different context or timeline is likely to produce similar results). In a further study, I suggest inserting an ultimate question, such as: My company is timely-efficient, and this efficiency allows making more money (or something similar). As the efficiency of the manager's work is a challenging, broad topic and provides opportunities for different perspectives on its solution, we have chosen a certain procedure. Adequate and very beneficial comments from opponents and proposed procedures and clarifying approaches to solving the problem will be part of further research.

The ultimate question must embrace some type of consequence or implication of a work-efficiency policy, that is, an indicator that reflects the effects of the strategic actions. Perhaps this question already exists in your questionnaire, but as you did not present it, I cannot propose anything. This ultimate question operates as an endogenous, dependent variable and forms a regression model (y= f(x1, x2,… xn)) with the current exogenous, independent ones. It would greatly increase the impact of your study if such a question exists because it could give a clue on what practice, belief, or behavior is more effective in increasing time productivity. The research was the first of the implemented and based on it, we will apply the proposed procedures based on your comments.

The analysis and discussion is fine. In the last chapter, please avoid lists and mainly lists within lists. This is a closing chapter in which you finish the article and provide a clue on further research. A list of results and a discussion on their implications should be located in the previous chapter. Congrats on your study. Looking forward to receiving your next version. DONE

Reviewer 2 Report

The paper fits within the scope of the journal. Research of a large extent was performed. 

However, its global scientific soundness is doubtful. The presented research of a particular sector (food companies) in particular countries is more like a case study. Are conclusions characteristic only to food industry? The authors should “envelope” the research by describing scientific aims of the research and its contribution to global knowledge.

Specific comments:

1) Abstract is too long; it should be more concise.

 2) Please add a paragraph at the beginning of introduction that introduces and describes the problem under research. Introduction is too long. Maybe it is worth to split it into Introduction and Literature review?

3) The newest references are missing. Please refer to the recent literature (2020-2021) to show that the authors are familiar with up-to-date state of the research.

4) Line 408-409. “… marked in yellow…” There is nothing in yellow in the Table.

5) Non-English words in legends of figures appear (“po 18:00”).

6) ICT, IOR – please, explain the acronyms when they are used in the text for the first time.  

Author Response

Are conclusions characteristic only to food industry? Our conclusions are characteristic not only for the food industry. We are currently conducting research in companies from other sectors of the national economy. The findings processed so far have confirmed similar results.

The authors should “envelope” the research by describing scientific aims of the research and its contribution to global knowledge.  DONE line 89-95

Specific comments:

1) Abstract is too long; it should be more concise. DONE

 2) Please add a paragraph at the beginning of introduction that introduces and describes the problem under research. Introduction is too long. Maybe it is worth to split it into Introduction and Literature review? „Introduction“ and „Stating the problem“.

3) The newest references are missing. Please refer to the recent literature (2020-2021) to show that the authors are familiar with up-to-date state of the research. DONE

4) Line 408-409. “… marked in yellow…” There is nothing in yellow in the Table. DONE

5) Non-English words in legends of figures appear (“po 18:00”). DONE

6) ICT, IOR – please, explain the acronyms when they are used in the text for the first time. DONE 

Round 2

Reviewer 1 Report

Authors have satisfactorily addressed most issues